# Unemployment Syndrome during COVID-19: A Comparison of Three Population Groups

**DOI:** 10.3390/ijerph18147372

**Published:** 2021-07-09

**Authors:** Anna Bocchino, Ester Gilart, Inmaculada Cabrera Roman, Isabel Lepiani

**Affiliations:** 1Nursing University Salus Infirmorum of Cadiz, 11001 Cadiz, Spain; Isabel.lepiani@ca.uca.es; 2Lopez Cano Hospital of Cadiz, 11010 Cadiz, Spain; esther.gilart@gmail.com; 3Emergency Department Bahia de Cadiz, University of Cadiz, La Janda, 11001 Cadiz, Spain; valentinaspain@hotmail.com

**Keywords:** unemployed syndrome scale, COVID-19, symptoms

## Abstract

Introduction: Of the serious problems that characterise the current crisis in Spain, the most alarming and revealing is unemployment, which, despite being so common, continues to be quite a negative experience for most people, often with serious negative effects on their biopsychosocial health. The perpetuation of this situation has given rise to a new syndrome of the unemployed. If these effects of economic downsizing are accompanied by the magnitude of the current situation brought about by COVID-19, the results can be devastating for the individuals and families experiencing it. Objective: To compare the symptoms of the unemployed syndrome in three population groups. Method: Three groups were studied: short-term unemployed (*n* = 91), long-term unemployed (*n* = 150), and those unemployed during the COVID-19 pandemic (*n* = 94). Unemployment syndrome was assessed with the Unemployment Syndrome Scale (USS). The three population groups were contacted through web pages, social networks, etc. and answered the instruments online in a single session. Once the responses were obtained, their information was encoded in a database and analysed through the SPSS v. 21 program. Population groups were compared using the ANOVA analysis and the Bonferroni post hoc test. Results: The unemployed individuals who lost their job during the pandemic reported higher scores in the symptoms of the Unemployed Syndrome Scale compared to the long- and short-term unemployed individuals. ANOVA analyses for symptoms of USS were all significant in the different groups considering a significance level of <0.005. Participants who were unemployed for less than one year had lower scores in the USS than the long-term unemployed participants and those unemployed during the COVID-19 pandemic that reported a significantly higher number of symptoms in the USS.

## 1. Introduction

COVID-19 is an emerging infectious disease caused by the SARS-CoV-2 virus that was first reported on 31 December 2019 in Wuhan, China. The importance given to this infection lies in the fact that it has quickly become a public health problem affecting many countries around the world. According to data from the Health Alerts and Emergencies Coordination Centre (CCAES), updated as of April 2021, 3,291,394 confirmed cases of COVID-19 have been reported in Spanish territory, and 75,541 people have died with this disease—the regions of Madrid, Catalonia, and Andalusia were the most affected [1]. COVID-19 is not only a public health crisis—recently, numerous studies and analyses have emerged on the possible impacts and economic consequences produced by this global pandemic [2,3,4,5,6]. Some of these have focused on the labour market. In Spain, the direct and indirect effects of some necessary containment measures, given the rapid spread of the virus, have begun to manifest in a gradual increase in unemployment, as well as in a reduction in the wages and salaries of workers, abruptly altering the global economy. The general activity and the hotel, restaurant, transport, and leisure sectors have suffered a very pronounced drop [7,8]; depending on the geographical area, it was particularly intense in some regions of Southern Spain. The latest Labour Force Survey (EPA) reflects that the unemployment rate has risen to 16.3%, affecting 527,000 more workers than the previous year. The majority (39.4%) are long-term unemployed (more than two years in unemployment) [9]. This is the largest increase in unemployment since 2012, accumulating 3.72 million unemployed. All of this represents an unprecedented economic crisis that is having devastating effects on individuals who struggle every day to safeguard, as far as possible, the economic and social well-being of their own family. Thus, among the stressful conditions faced by the unemployed today we find the coronavirus disease (COVID-19).

However, the negative consequences associated with unemployment, both physically and psychosocially, do not affect all individuals in the same way; in fact, a series of variables such as gender, age, length of unemployment, or cultural and social context of unemployed people have been considered key factors, influencing the psychological and social situation of people who face this process of economic reduction or deprivation. Different studies have shown how families with unemployed children and parents face a higher level of difficulty, with possible long-term consequences for the well-being and development of the child [10,11,12,13]. If the magnitude of these economic reduction effects is significant, it can lead to a series of devastating results for individuals and families who experience these effects to such a degree that they are able to speak of a new unemployment syndrome [14]. Recent research has shown how the unemployed syndrome has a series of negative repercussions on the health of people who suffer from it. The symptoms are broad and encompass both physical factors (tachycardia, digestive disorders, etc.) and psychological factors (anxiety, depression, anger, frustration, fear, etc.), as well as behavioural factors (alcohol and drug abuse, sleep disorders, etc.) [15].

Even though anger, depression, aggressiveness, anxiety, frustration, etc. are among the most evident symptoms, which by themselves represent an alarming concern, as recent research has been able to show [16] and which also provide several reasons to make long-term unemployment a plausible topic worthy of attention, the current crisis has made us wonder why conceptual, empirical, and clinical knowledge on this topic is currently very limited. Assessing the negative impacts of unemployment is fraught with challenges. First, to date, there is only one recently validated diagnostic instrument [16], and its usefulness in clinical practice is still partly unknown. Secondly, health professionals are neither prepared nor trained to care for this type of population since they are unaware of the possible symptoms related to the unemployment syndrome. In this way, although many authors have focused their efforts on studying the different factors related to unemployment, as well as the main associated consequences [17,18,19,20,21], there are still no unambiguous results regarding the impact that years of unemployment can have on people, and there is even less knowledge on the negative consequences of unemployment caused by the current COVID-19 situation which we are facing. Therefore, this study analysed the unemployment syndrome in three different situations: short-term, long-term, and caused by the pandemic effect, in order to estimate its prevalence and develop appropriate measures for its treatment. On the basis of the aforementioned discussion, we formulated the following hypotheses. Our first hypothesis is that there will be significant differences between the short-term, long-term, and COVID-19-related unemployed with respect to the scores obtained on the Unemployed Syndrome Scale (USS) (H1).

Our second hypothesis (H2) is that the negative physical consequences or symptomatology of the unemployed will be aggravated depending on the unemployment period—the longer the time spent unemployed, the greater the physical deterioration. Finally, according to our last hypothesis (H3), we hope the unemployed that have lost their job during the current COVID-19 crisis obtain a higher score regarding psychological symptoms.

## 2. Materials and Methods

### 2.1. Participants

A non-probabilistic convenience sampling was carried out. The final sample of voluntary participants had 335 people, with an age range between 19 and 60 years old. Regarding gender, 62.1% were men and 37.9% were women. The sample was divided into three population subgroups: (1) short-term unemployed (less than 2 years), (2) long-term unemployed (more than two years), and (3) unemployed individuals who lost their job during the COVID-19 pandemic. In the latter group, there was a filter question asking if they had lost their job due to COVID-19, in order to avoid possible confusion with the first subgroup.

### 2.2. Measures

#### 2.2.1. Demographic Characteristics

A questionnaire was used to collect demographic and socio-economic data considered useful for the purposes of the study, such as age, gender, unemployment period, number of children, academic level, and economic benefits.

#### 2.2.2. Unemployed Syndrome Questionnaire

The version of the USS measure tested by Bocchino, Lepiani Diaz, Gilart Cantizano, Medialdea and Dueñas Rodríguez [16] is comprised of 20 items related to symptoms caused by unemployment conditions.

All items were rated on a 1 to 5 scale with responses ranging from always to never. The final USS score is the sum of the scores of the items. The summed USS scores can range from 20 to 100. A high score on the scale reflects a high level of unemployment syndrome. Using the visual grouping method, 4 severity categories were created (1: mild; 2: moderate; 3: severe; 4: extremely severe). The scale presents a factorial structure of three factors: Dimension 1 is related to the psychic/cognitive aspect, dimension 2 is related to the physical and/or somatic symptoms, and dimension 3 is related to the social and behavioural aspects.

### 2.3. Recruitment and Procedures

A quantitative cross-sectional study was conducted in the form of an online survey using SurveyMonkey software. Three main mechanisms were used to alert potential participants: (1) advertising on Facebook and Twitter, (2) government and academic websites (e.g., Andalusian Employment Service), and (3) invitations to newsgroups and mailing lists.

Voluntary participants followed a link to the survey where they were presented with a participant information sheet and a consent form. If they agreed to continue with the survey, they automatically signed the informed consent. A screening question requiring participants to answer whether they had lost their job because of the pandemic crisis. Our data collection time points span from April 2020 to March 2021.

The study was conducted in agreement with the Declaration of Helsinki. The researchers only had access to anonymised data. The participants received no rewards for their participation.

Once the responses were obtained, their information was encoded in a database and analysed through the SPSS v. 21 program.

### 2.4. Statistical Analysis

A descriptive analysis of the sample was carried out using percentages for the qualitative variables and means, medians, and standard deviation (sSD for the quantitative variables. A comparative analysis of three groups of unemployed individuals was performed: short-term unemployed, long-term unemployed, and unemployed individuals who lost their job during the COVID-19 pandemic. An analysis of variance (ANOVA) was performed to compare the means of the three groups with the socio-demographic variables and the scores obtained in the items of the scale, using the Bonferroni post hoc test to identify the groups in which the significant differences were found. An alpha risk of 0.05 was considered.

Analyses were conducted with the statistical package SPSS 21.0 (IBM Corp., Armonk, NY, USA).

## 3. Results

The total number of the sample was 335 unemployed (208 men and 127 women) with an age range between 19 and 60 years old.

Demographic information (Table 1) indicated that the sample was characterised by 335 people: 91 short-term unemployed, 150 long-term unemployed, and 94 unemployed individuals who lost their job during the COVID-19 pandemic.

The majority of participants were men (62.1%) with a mean age of 41.8 years, who reached secondary education (43.9%) and had 2–3 children. The majority were not receiving economic benefits (55.8%). The total score of the scale indicated that the majority of the participants suffered from severe conditions of unemployed syndrome.

Descriptive statistics of the sample by employment status are provided in the same table. It is possible to observe significant differences among the three groups with respect to marital status, educational level, age, income, and the total score of the scale. The number of children, gender, and being responsible for the household income were not significant. For the variables of gender, being responsible for the household income, and number of children, the results showed that a higher proportion in all three groups of unemployed people were male, most of whom were responsible for the household income and most of them having 2–3 children. Regarding age, the majority of the long-term unemployed were in the fifties or older age bracket, while a majority of the participants unemployed during the COVID-19 period and short-term unemployed were in the 40–50 age bracket (*p* < 0.005). In addition, in the three groups, the majority of the participants were people who reached secondary education, followed by participants with university education. Finally, the majority of the sample declared that they do not receive other economic benefits (*p* < 0.005).

Regarding the general indices, in relation to the global severity index, significant differences were observed among the groups. Most of the participants reported a severe level of unemployed syndrome. The long-term unemployed were undoubtedly the hardest hit.

The reliability of the instrument was verified through Cronbach’s alpha. Internal consistency of USS was excellent, with Cronbach’s alpha value of 0.94.

ANOVA analysis for symptoms of USS were all significant in the different groups, considering a significance level of *p* < 0.005 (Table 2).

Participants who were unemployed for less than one year had lower scores on the USS than the long-term unemployed and unemployed during the COVID-19 pandemic, who reported a significantly higher number of symptoms in the USS.

Particularly in the group of individuals unemployed during the pandemic, a greater symptomatology of the unemployment syndrome was observed, namely, stress (4.03), deterioration of the quality of life (4.00), anxiety (4.27), hostility (4.06), impotence (4.29), frustration (4.33), sleep pattern disorders (4.05), fear (4.24), feeling of irritability (4.19), and lack of adaptive resources and management of stressor (3.95), having a greater intensity compared to the other two groups (Table 3). The same group obtained higher scores in cardiovascular symptoms (4.29); abuse of alcohol, tobacco, and other harmful substances (3.05); and ineffective coping strategies (4.03). In contrast, the long-term unemployed showed higher scores only in low self-esteem (4.02), depression (3.58), hopelessness (4.05), and low personal satisfaction (4.11).

Post hoc analysis with the Bonferroni adjustment showed that there were significant differences with respect to the total score of the scale in the three groups (F_2; 330_ = 34.092; *p* = 0.000). With regard to symptomatology, the short-term unemployed showed significantly different scores in almost all symptoms compared to the long-term unemployed and the unemployed who had lost their job during the COVID-19 pandemic. However, exceptions were stress, sleep pattern disorder, feelings of irritability, and lack of strategies for coping—a difference that was clear in the three groups, with the individuals unemployed due to COVID-19 being the most affected. The hostility symptom again showed a significant difference between those unemployed during the pandemic and the other groups. With regard to hopelessness, the long-term unemployed had a significant difference with respect to the other groups.

## 4. Discussion

The impact of COVID-19 and its bio-psycho-social consequences is fraught with challenges for the general public and, in terms of labour, has been particularly difficult for unemployed people, who often face great physical and psychological overload in highly stressful conditions from the very beginning.

Although in recent research, the pandemic has shown its devastating impact on different populations [22,23,24,25,26,27,28,29,30], thus far, there are no previous studies that have considered the impact on mental health in unemployed people and their families, especially in the Spanish context. Thus, this study aimed to evaluate the differences in the scores obtained in the unemployed syndrome scale considering the unemployment period or the cause of unemployment. The sample consisted of three population groups: short-term unemployed, long-term unemployed, and those unemployed during the COVID-19 pandemic.

At the beginning of the study, three hypotheses were raised.In the first hypothesis (H1), it was stated that there would be significant differences between the three groups with respect to the symptoms of USS. The results obtained showed significant differences not only in the total score of the scale, but also in each of the symptoms—corroborating our first hypothesis. However, as expected and according to previous studies [31,32], these differences were more evident between the short-term unemployed and both the long-term unemployed and those unemployed during the pandemic.

Our second hypothesis (H2) stated that the negative physical consequences of the unemployed would be aggravated depending on the unemployment period, while in our third hypothesis (H3), it was expected that the unemployed who had lost their job during the COVID-19 crisis would have obtained a higher score in psychological symptoms. In response to these claims, we found the most interesting finding of the study: Although those who were unemployed due to COVID-19 had lost their job for a relatively short period of time and were comparable to the short-term unemployed, they showed both physical and psychological symptoms for the most part. These were so high that the Bonferroni test did not indicate significant differences between the long-term unemployed and those unemployed due to the pandemic. In these two groups, there were no significant differences with respect to the dimensions and the total score of the scale. However, there were differences when comparing the results with the short-term unemployed. This finding, although not in line with other research on unemployment that states that health tends to progressively worsen as the length of the unemployment period increases [33,34,35], could be explained by referring to the cognitive processes involved in losing a job. In other words, the causal attribution of unemployment could be associated with individual, social, or fatalistic causes and could influence the bio-psycho-social responses of the individual. In other words, the attribution of unemployment to factors not related to a person (such as a pandemic event) could be associated with negative responses or consequences on an individual’s health. To this extent, research has made reference to the fact that people who experience involuntary unemployment or whose unemployed condition persists may be a targeted group who need preventive mental health services [36,37,38,39].

However, notwithstanding the data obtained, it should be noted that the greater participation of long-term unemployed individuals should lead us to take the results and the differences obtained with some caution. Likewise, the generalisation of the results is limited since it is a non-probabilistic sample in which there may be a certain selection bias: participation was voluntary and those especially impacted by unemployment may have participated. Future studies should expand the sample, obtaining a more balanced non-probabilistic sample with respect to the three groups and extend it to different regions.

## 5. Conclusions

To conclude, it is necessary to underline the fact that the unemployed syndrome is very present among the sample that participated in the study. Moreover, this finding is even more evident in people who have lost their jobs during the pandemic. COVID-19 is currently experienced as an unexpected and stressful factor, largely due to the great knowledge gap that still exists on this issue, which arouses feelings of vulnerability or helplessness and affects both personal and family health, as well as the labour market, which in turn is full of uncertainty.

Thus far, there are no studies in Spain that explore the syndrome of the unemployed during the pandemic. This would be the main strength of the study. The results presented in this manuscript have been offered as a means to stimulate future research, the development of programs for unemployed persons, and advocacy efforts. Naturally, these results are not intended to be exhaustive. They are intended to provide evidence of the symptoms that people with unemployment syndrome may experience in order to stimulate the active pursuit of transformative interventions and policies needed for individuals and communities experiencing job loss (including people who have lost their jobs during the pandemic). This research is intended as a starting point for future research that includes other types of qualitative and participatory methodology in order to channel strategies from the lived experiences of individuals who are now out of work. We also advocate the use of the USS scale in other populations or contexts to assess symptoms of unemployment and to develop a new understanding of the nature of unemployment in order to develop and evaluate specific, individualised interventions. From a clinical point of view, there is a need for a self-administered measure that reflects the current definition of unemployment syndromes and also meets some criteria such as patient acceptability, simplicity, low cost, and psychometric validity [16]. Moreover, the USS is a measure that allows for the quantification of the symptoms of the main indicators of a syndrome in a short period time and can be of great use for both researchers and clinicians working in the healthcare field (e.g., as an aid for screening and clinical diagnosis, to quantify the level of disorder, or to evaluate the effectiveness of treatment). Finally, and starting from the premises that both the unemployed syndrome and the pandemic have consequences for mental health and psychosocial well-being, from a more global perspective, policymakers have the opportunity to mitigate the negative effects of unemployment in times of COVID-19 by ensuring, to the extent possible, family income support for families in need. They should also ensure that a wide range of activities are put in place to protect or promote psychosocial well-being or prevent or treat mental health conditions. For example, it could be useful to develop and expand appropriate public and subsidised services, such as free online counselling, support groups that provide emotional and practical support, and support for families in need.

## Figures and Tables

**Table 1 ijerph-18-07372-t001:** Characteristics of the sample and descriptive statistics by employment status.

		Period Unemployment			Total *n* (%)	
		Short-Term	Long-Term	Covid Period		*p*-Value
Sex	Man	56	93	59	208 (67.1)	
	Women	35	57	35	127 (37.9)	0.985
Marital status	Single	9	2	6	17 (5.1)	
	Partner relationship	19	15	13	47 (14.0)	
	Married	37	66	19	122 (36.3)	
	Separated	20	45	50	115 (34.3)	0.000
	Divorced	6	11	3	20 (6.0)	
	Widower	0	11	3	14 (4.2)	
Age	29–39	4	9	8	21 (6.3)	
	40–50	47	53	50	150 (44.8)	0.016
	51–61	40	88	36	164 (49.0)	
Head of the income	No	24	36	15	75 (22.4)	
	Yes	67	114	79	260 (77.6)	0.193
Income	No	50	95	42	187 (55.8)	
	Yes	41	55	52	148 (44.2)	0.017
Educational level	Without studies	17	9	18	44 (13.1)	
	Primary school education	10	17	8	35 (10.4)	
	Secondary school education	32	72	43	147 (43.9)	0.023
	Higher education	32	52	25	109 (32.5)	
Children	0–1	30	41	26	97 (29.0)	
	2–3	59	92	58	209 (62.2)	0.166
	<3	2	16	10	28 (8.4)	
USS total	Mild	16	1	0	17 (5.1)	0.000
	Moderate	14	5	3	22 (6.6)	
	Severe	47	87	55	189 (56.8)	
	Extremely severe	14	55	36	105 (31.5)	

USS—Unemployment Syndrome Scale.

**Table 2 ijerph-18-07372-t002:** Means differences for USS symptoms.

Symptoms	Short-Term Period	Long-Term Period	COVID Period	*p*-Value *
	Mean (SD)	Mean (SD)	Mean (SD)	
Stress	3.16 ^a^ (1.276)	3.69 ^b^ (0.636)	4.03 ^c^ (0.663)	<0.01
Endocrine symptoms	2.82 ^a^ (0.961)	3.21 ^b^ (0.782)	3.32 ^b^ (0.882)	<0.01
Deterioration of the quality of life	3.20 ^a^ (1.147)	3.97 ^b^ (0.838)	4.00 ^b^ (0.842)	<0.01
Low self-esteem	3.46 ^a^ (1.109)	4.02 ^b^ (0.807)	3.94 ^b^ (0.853)	<0.01
Anxiety	3.60 ^a^ (1.104)	4.00 ^b^ (0.843)	4.27 ^b^ (0.721)	<0.01
Depression	2.97 ^a^ (0.936)	3.58 ^b^ (1.018)	3.38 ^b^ (0.985)	<0.01
Hostility	3.34 ^a^ (1.035)	3.61 ^a^ (1.022)	4.06 ^b^ (0.865)	<0.01
Apathy	3.18 ^a^ (1.091)	3.69 ^b^ (0.955)	3.64 ^b^ (0.926)	<0.01
Hopelessness	3.20 ^a^ (1.166)	4.05 ^b^ (0.792)	3.52 ^a^ (1.233)	<0.01
Fear	3.45 ^a^ (1.176)	4.10 ^b^ (0.873)	4.24 ^b^ (0.838)	<0.01
Feeling of powerlessness	3.59 ^a^ (1.115)	4.23 ^b^ (0.806)	4.29 ^b^ (0.785)	<0.01
Gastrointestinal symptoms	2.80 ^a^ (0.734)	3.13 ^b^ (0.730)	3.13 ^b^ (0.765)	<0.01
Feeling of irritability	3.23 ^a^ (1.116)	3.60 ^b^ (1.003)	4.19 ^c^ (0.919)	<0.01
Frustration	3.45 ^a^ (1.241)	4.10 ^b^ (0.841)	4.33 ^b^ (0.781)	<0.01
Low personal satisfaction	3.42 ^a^ (1.350)	4.11 ^b^ (0.840)	4.10 ^b^ (0.763)	<0.01
Sleep pattern disorders	3.29 ^a^ (1.293)	3.69 ^b^ (0.851)	4.05 ^c^ (0.932)	<0.01
Abuse of alcohol, tobacco, and other harmful substances	2.48 ^a^ (0.993)	2.95 ^b^ (1.012)	3.05 ^b^ (0.932)	<0.01
Cardiovascular symptoms	3.32 ^a^ (1.349)	4.04 ^b^ (0.881)	4.29 ^b^ (0.825)	<0.01
Lack of adaptive resources and management of stressor	3.19 ^a^ (1.316)	3.89 ^b^ (0.512)	3.95 ^b^ (0.494)	<0.01
Ineffective coping strategies	3.16 ^a^ (1.276)	3.69 ^b^ (0.636)	4.03 ^c^ (0.663)	<0.01

* Note: *p* < 0.01. Different letters in the means represent statistically significant differences (*p* ≤ 0.05) within the groups.

**Table 3 ijerph-18-07372-t003:** Mean differences of the dimensions of the scale and its total score in the three groups of unemployed individuals.

Dimensions **	Short-Term Period	Long-Term Period	COVID Period	*p*-Value *
	Mean (SD)	Mean (SD)	Mean (SD)	
Factor 1	36.61 ^a^ (10.17)	43.67 ^b^ (7.07)	43.85 ^b^ (6.11)	<0.01
Factor 2	12.23 ^a^ (3.44)	14.08 ^b^ (2.17)	14.78 ^b^ (2.34)	<0.01
Factor 3	12.03 ^a^ (3.95)	14.49 ^b^ (1.93)	15.03 ^b^ (2.06)	<0.01
USS total	64.10 ^a^ (17.71)	75.95 ^b^ (10.18)	77.86 ^b^ (9.41)	<0.01

** Factor 1 represents to the psychic/cognitive dimension of USS. Factor 2 represents the physical and/or somatic dimension of USS. Factor 3 represents the social and behavioural dimension of USS. * *p* < 0.01. Different letters in the means represent statistically significant differences (*p* ≤ 0.05) within the groups. USS—Unemployment Syndrome Scale.

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
