# Peer review of "Unemployment Syndrome during COVID-19: A Comparison of Three Population Groups"

_ijerph, 2021, doi:10.3390/ijerph18147372_

Round 1

Reviewer 1 Report

The authors report results from a study investigating unemployment among three population groups: a first group of short-term unemployed individuals, a second group of long-term unemployed, and a third group of individuals unemployed during the COVID-Pandemic.

To evaluate specific symptomatology associated with unemployment, they used the Unemployment Syndrome Scale. The paper is of interest for the journal, however; before publishing it, I would recommend some changes.

In the abstract section, the authors have omitted a little introduction about unemployment syndrome. I recommend to introduce the relevance of the syndrome in the introduction section.

In the abstract, the authors do not describe the methods they used for the recruitment of individuals, and how they evaluated items of the USS scale. Results are described in the abstract in a general way. I would recommend to describe "findings" instead of analyses in the results section. 

In the introduction section, the authors firstly introduce the topic of COVID, as an emerging disease. The paper is based on the unemployment, and they evaluate three groups. I would prefer to introduce the topic of unemployment first, and afterwards, what is associated with the COVID-19 pandemic.

In the methods section, the authors describe that a questionnaire was used to collect information. How was the questionnaire administered? They should better clarify this point.

In the conclusion section, I would extend the paragraph about clinical perspectives, by integrating future implications for research and for planning social and mental health programs taking into account these findings.

Reviewer 2 Report

Dear Authors!

Thank you for taking up a very interesting and important topic of Unemployment Syndrome During Covid-19.

Despite the selection of an interesting topic, the overall quality of the text is poor. Below are just a few of the many issues that need to be improved:

  • "Nursing" is not an appropriate keyword in the context of the content of this article.
  • The description of the sample selection is not exhaustive. I can only guess that it was a non-random, deliberate sampling.
  • Based on which source did the authors assume that short-term unemployment is less than 2 years with no job? It is worth mentioning such a source.
  • The text is not sufficiently grounded in theory (the authors only refer to 13 sources in total).
  • Reference to hypotheses should be made in the "Result" section and not in the "Discussion".
  • The research results were not properly discussed.
  • The authors did not present the implications of the study results.
  • There are also other errors and inaccuracies, for example in the "Materials and Methods" section the authors claim that the age of the respondents is between 18 and 60 years old. However, in the "Results", they declare the range 19-60 years old.

Therefore, I cannot recommend the article for publishing in a journal with a reputation as IJERPH.

Best regards,
The reviewer.

Reviewer 3 Report

Thank you for the opportunity to review.  A couple comments to help improve the manuscript:

  • the intro discusses employment in Spain as related to COVID, but does not clearly address employment trends in this region at the short-term and long-term prior to the pandemic.
  • it would be very helpful to clearly distinguish between short-term and during COVID unemployment periods earlier in the paper.  It's mentioned in the methods, but I was confused on these time periods up until that part of the manuscript.
  • results:  if SPSS was used, why not simply use the ANOVA output tables from the software?  I'm not following tables 2 and 3 - there's no sum of squares, DF, or F test results for each variable?  I'm assuming M is mean, but not sure what DT represents?  DT is not clearly explained in the section.
  • please clearly label/explain the 3 groups (labeled 'factors') in table 2.

I think if the results clearly resembled that of SPSS output tables I would be able to follow much better.  Or, explain what DT is in the table.

With p-value significance findings as they are across all variables - is there a suggested (better) statistical test that may be available for future study?  One may assume that USS findings during periods of unemployment and especially during COVID would result in higher prevalence of symptoms.  What can be posited from these p-values and areas for future study (besides replicating the study outside of Spain)?

Thank you

Round 2

Reviewer 2 Report

Dear Authors!

Thank you for referring to my comments as they had a positive effect on the quality of the article. Unfortunately, not all of the changes you have made can be considered sufficient:

  • The description of the sample selection still needs to be clarified. Were the respondents selected randomly or by purposive sampling or maybe by convenience sampling? How was the study conducted? how did you reach the respondents?
  • The text is still not sufficiently grounded in theory and related to other studies. You only added a few references. I cannot agree with your conclusion that “The contribution of a small number of references is due to the novel and unprecedented subject matter”. There have been many articles recently dealing with similar issues. After typing in google scholar: "covid" and "unemployment" 60,000 papers appear.

Therefore, I still cannot recommend the article for publishing in a journal with a reputation as IJEREPH.

Best regards,

The reviewer.
